# Quantifying the benefit of a proteome reserve in fluctuating environments

Matteo Mori [1], Severin Schink [1,2], David W. Erickson[1], Ulrich Gerland[2] & Terence Hwa [1]

The overexpression of proteins is a major burden for fast-growing bacteria. Paradoxically, recent characterization of the proteome of *Escherichia coli* found many proteins expressed in excess of what appears to be optimal for exponential growth. Here, we quantitatively investigate the possibility that this overexpression constitutes a strategic reserve kept by starving cells to quickly meet demand upon sudden improvement in growth conditions. For cells exposed to repeated famine-and-feast cycles, we derive a simple relation between the duration of feast and the allocation of the ribosomal protein reserve to maximize the overall gain in biomass during the feast.

[1] Department of Physics, University of California at San Diego, 9500 Gilman Drive, La Jolla, CA 92093-0374, USA. [2] Physics Department, Physics of Complex Biosystems, Technical University of Munich, James-Franck-Str. 1, D-85748 Garching, Germany. Matteo Mori, Severin Schink, and David Erickson contributed equally to this work. Correspondence and requests for materials should be addressed to T.H. (email: hwa@ucsd.edu)

Exponentially growing bacteria adapt their proteome composition to the medium they grow in[1–7]. Recent research has established that the coarse-grained characteristics of proteome composition varies mostly with the growth rate of the culture according to the mode of growth limitation (e.g., carbon limitation vs antibiotic inhibition), and is insensitive to the details of the growth condition[4, 7]. A prominent example of the coarse-grained proteome sectors is the ribosome-affiliated "R-sector", which includes ribosomes and the affiliated translation machinery, collectively referred to as R-proteins, and is responsible for protein synthesis. Its "proteome fraction" $\phi_R$ is defined as the total mass of R-proteins $M_R$ per total protein mass $M$, i.e., $\phi_R \equiv M_R/M$. Since the abundance of total proteins per cell volume is constant in the conditions we are interested in (Supplementary Fig. 1), protein mass fractions can be considered equivalent to protein concentrations[8, 9].

For different nutrients giving different steady state growth rates $\lambda^*$, the R-sector proteins occupy different shares of the proteome through the "growth law"[2–4, 7] (Supplementary Fig. 2a).

$$\phi_R^*(\lambda^*) = \phi_{R0} + \lambda^*/\nu_R. \tag{1}$$

(Here and below, steady state quantities are indicated by asterisks, to distinguish them from dynamic variables; the full list of symbols used in the text can be found in Supplementary Table 1.) The inverse slope $\nu_R$ of this linear relationship, when converted to appropriate units, compares well with the maximum in vitro elongation rate by the ribosomes, and is referred to as the "translational capacity" of the cell[4]. In fact, if every ribosome were engaged in translation at the maximal rate, one would expect a protein synthesis flux $\lambda^* M = \nu_R M_R$. Thus, $\phi_{R,min}^* = \lambda^*/\nu_R$ is the minimal ribosomal fraction needed to support exponential cell growth at a rate $\lambda^*$. Comparison of the growth law (Eq. (1)) with this minimal demand shows that the R-sector proteins are expressed in excess by an amount $\phi_{R0}$, i.e., an overcapacity, which equals the $y$-intercept of the growth law (Eq. (1)). Mechanistically, the cell achieves this overcapacity by an inactivation of ribosomes, which decreases the amount of active ribosomes, while keeping the elongation rate of the active ribosomes reasonably high[10]; indeed, during slow growth as much as 80% of all ribosomes are in a non-translating state[10]. This high degree of overcapacity raises the obvious question of what benefit this overcapacity might provide for the cell. It becomes even more

puzzling considering that the overexpression of useless proteins reduces growth rate by a proportional amount, as has been validated quantitatively for specific overexpression systems[4]. Since $\phi_{R0}$ represents an amount of over-expressed R-proteins, it is a burden for the cell during steady state growth.

The overcapacity of ribosomes has been noted in the past[8, 11–14], and it was speculated that this it is an investment which becomes advantageous during growth upshift[11, 15–17]. Adaptation following abrupt changes in nutrient conditions, e.g., where a good nutrient source is added to the growth medium during exponential growth on a poor substrate, is conveniently studied in the laboratory[2, 5, 18–20] and occurs widely in natural ecological context[21]. In this work, we quantitatively connect the overcapacities in R-sector to the famine-to-feast transition encountered in natural environments such as the mammalian gut: Rich nutrients are provided for a limited period of time ("feast time"), e.g., a few hours following meals, and quickly exhausted outside of this time window. We develop a theory to capture the growth transition kinetics and predict the extent by which growth speeds up during upshift for different amount of R-sector overcapacity. The predicted adaptation dynamics is probed experimentally, and the predicted dependence on the pre-shift R-sector content is validated by varying pre-shift growth. Since the R-sector speeds up growth during transition while slowing down growth in the long run, we interpret them as a "reserve", kept by the cell to meet increased demand more quickly in an improved environment. Our theory allows us to compute the amount of this reserve that is expected to be selected evolutionarily because it maximizes the overall biomass accumulated throughout the course of the shift, for different duration of feast time and growth medium.

## Results

**Upshift kinetics and translational efficiency.** In this section we present and analyze a simple kinetic model that connects growth transition kinetics to ribosomal protein overcapacity in a famine-and-feast scenario (Fig. 1a). We assume that the addition of rich nutrients provides the cell with saturating amounts of building blocks (e.g. amino acids and nucleotides), such that cell growth is limited by the capacity to synthesize proteins, rather than by the metabolic and biosynthetic capacity. This assumption is well supported by the results of a series of nutrient upshift experiments from extremely slow growing cells (doubling time longer

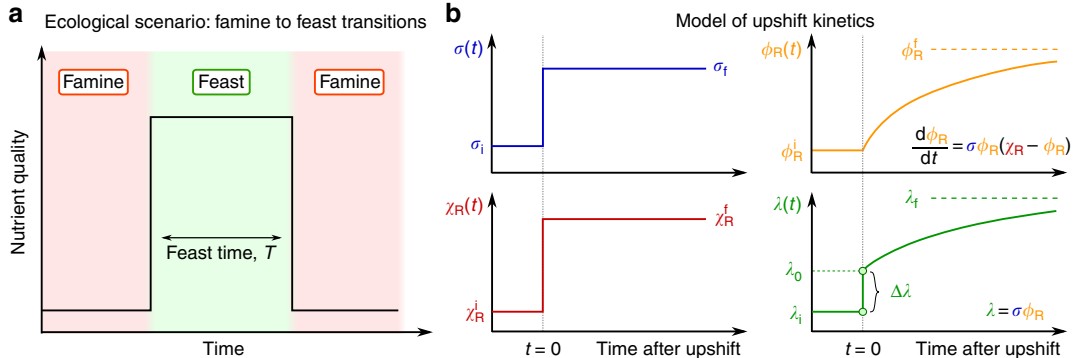

**Fig. 1** Upshift kinetics. **a** Famine-to-feast transitions. As rich nutrients are introduced into the environment, bacterial cells undergo a nutritional shift from a poor to rich medium, speeding up their growth; then, nutrients get depleted after a "feast time" $T$. If the feast time is short, repeated transitions between famine and feast will select for quickly adapting cells. **b** Kinetics of an upshift from poor to rich nutrients. Protein synthesis depends on two quantities, the (average) translational efficiency $\sigma(t)$ and the fraction $\chi_R(t)$ of newly produced proteins which are R-proteins. Knowledge of these two quantities as a function of time is sufficient to completely determine the kinetics, including the R-protein mass fraction $\phi_R(t)$ (through Eq. (5)) and the growth rate $\lambda(t) = \sigma(t)\phi_R(t)$. At time $t = 0$, both the translational efficiency and the R-proteins synthesis flux shift from the initial ($\sigma_i$ and $\chi_R^i$) to their final, post shift, values ($\sigma_f$ and $\chi_R^f$). The mass fraction of R-proteins $\phi_R(t)$ adjusts slowly toward its final value $\phi_R^f = \chi_R^f$. The growth rate (Eq. (6)) has quick jump $\Delta\lambda$ from the initial $\lambda_i$ to a larger value $\lambda_0$, due to the increase in translational efficiency, followed by a slower convergence to the new steady state growth rate $\lambda_f$ due to the slow increase in ribosome concentration. The value of the growth rate at the shift, $\lambda_0$, depends on the abundance of ribosomal proteins before the shift (Eq. (8))

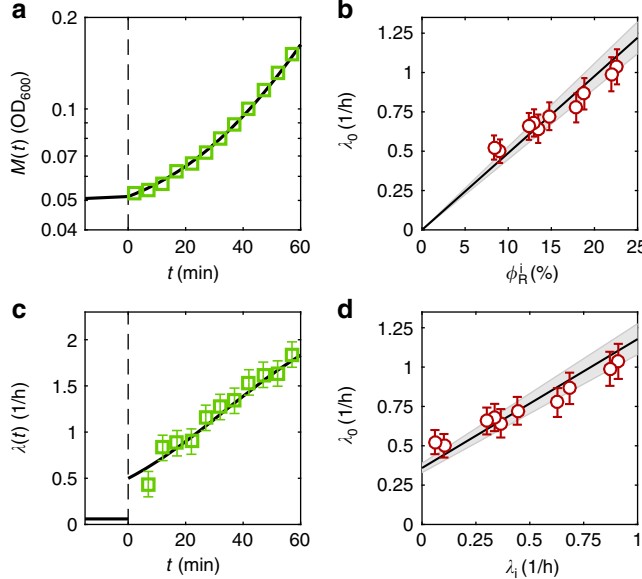

**Fig. 2** The kinetics in upshifts to rich media depend on the pre-shift ribosomal content. **a** Example of a shift from poor to rich media. Cells in exponential phase in carbon minimal media (aspartate, $\lambda_i = 0.06/h$) undergo a nutrient shift to rich media (LB + glucose, $\lambda_f = 2.45/h$) at time $t = 0$ (dashed line). The accumulation of protein mass $M(t)$ is measured by optical density (OD) at 600 nm (shown in log scale). Prior to the shift, mass accumulates as $M(t) \propto \exp(\lambda_i t)$ (data not shown); after upshift, the solid lines are given by Eq. (7), with $\lambda_0$ being the sole fitting parameter (the final growth rate is set to $\lambda_f = 2.45/h$). **b** Instantaneous growth rate for the growth transition shown in (**a**). Error bars (s.e.m.) are computed from the measurement uncertainty of $OD_{600}$ as described in Supplementary Fig. 4. After upshift, $t \geq 0$, the instantaneous growth rate increases according to Eq. (6) (solid line). **c** Experimental values of $\lambda_0$ (red circles, data in Supplementary Table 2. Error bars indicate fit uncertainties of $\lambda_0$ (s.e.m.)) are extracted from 10 independent upshifts from minimal media with different carbon sources to the same post-shift media (one upshift is shown in (**a**, **b**), other upshifts in Supplementary Figs. 3 and 4) and plotted against the pre-shift R-sector mass fraction, $\phi_R^i$ (absolute uncertainty: 1%, error bar not shown). The black line is the theoretical prediction from Eq. (8), the slope equals $\lambda_f / \phi_R^f = (4.9 \pm 0.4)/h$, based on independent steady state measurements of $\lambda_f = 2.45 \pm 0.2/h$ and $\phi_{R,f} = 0.50 \pm 0.01$. The shaded area indicates the uncertainty. **d** Experimental values of $\lambda_0$ (red circles) and plotted against the pre-shift growth rate, $\lambda_i$. Error bars indicate fit uncertainties of $\lambda_0$ (s.e.m.). The theoretical prediction is computed by inserting the R-line, Eq. (1), into Eq. (8)

than 10 h) to rich nutrient broth, where Koch et al. found the average protein synthesis rate to increase by several folds immediately after up-shift[15, 21]. Since the building blocks are not limiting after upshift, we model the total rate of protein synthesis $\dot{M}$ being proportional to the R-proteins mass $M_R$ as

$$\dot{M} = \sigma \cdot M_R. \tag{2}$$

Here, $\sigma$ is the "translational efficiency", which measures the average rate of protein synthesis per unit of R-protein mass; this quantity can also be expressed in terms of the number of actively elongating ribosomes and their elongation rate (see Eq. (4) in Supplementary Note 1). Expressing in term of $\phi_R \equiv M_R/M$, Eq. (2) becomes,

$$\dot{M} = \sigma \phi_R \cdot M. \tag{3}$$

In steady state, the mass fraction $\phi_R^*$ is set by the growth law, Eq.

(1). In changing environments, instead, $\phi_R(t)$ varies with time due to modulation in the rate of R-sector protein synthesis, $\dot{M}_R$, which is controlled by $\chi_R$, the fraction of total protein synthesis flux allocated to the R-proteins, i.e., $\dot{M}_R = \chi_R \dot{M}$. When combined with Eq. (2), we have

$$\dot{M}_R = \sigma \chi_R \cdot M_R, \tag{4}$$

which is a closed equation for $M_R$ subjected to the allocation $\chi_R$. The kinetics of $\phi_R(t)$ can be obtained by combining Eqs. (3) and (4), yielding a logistic equation

$$\dot{\phi}_R = \sigma \phi_R \cdot (\chi_R - \phi_R). \tag{5}$$

In balanced growth, where all cellular components increase at the same rate, we have $\chi_R^* = \phi_R^*$, with the latter given by the growth law Eq. (1). We focus here on the situation where cells experience a sudden shift from famine to feast, Fig. 1b. During the shift, cells transition between two different steady states: we will denote these states as "initial" and "final", with the corresponding quantities indicated by the labels "i" and "f", respectively. (Asterisks will indicate either of the two.) The cells are initially in balanced growth with growth rate $\lambda_i$, in a poor nutrient source. At the time of the upshift, rich nutrients are added and, as a consequence, both $\sigma$ and $\chi_R$ vary with time during growth transitions. A full treatment of the transition kinetics requires equations for $\sigma(t)$ and $\chi_R(t)$, and will be addressed elsewhere. Instead, for transitions to rich medium, it is known[12, 22] that the translational efficiency quickly increases from the initial value $\sigma_i$ to its final steady state value in the feast condition, $\sigma_f$. Also, the fractional rate of ribosome accumulation, $\dot{M}_R/M_R = \sigma \cdot \chi_R$ as given by Eq. (4), is found[12, 22, 23] to increase immediately to the post-shift value, thus implying that the R-protein synthesis fraction $\chi_R$ quickly shifts from $\chi_R^i$ to $\chi_R^f$. An increase in $\chi_R$ means that the synthesis of other protein sectors have to decrease, as all synthesis fractions $\chi_j$ have to sum up to unity (see Eq. (5) in Supplementary Note 1). Assuming instantaneous change of $\sigma$ and $\chi_R$ from their initial to final values (Fig. 1b), Eq. (5) can be solved analytically, and the R-sector protein fraction $\phi_R(t)$ increases toward its final value as a logistic function (see Eq. (12) in Supplementary Note 1). The instantaneous growth rate, $\lambda(t) \equiv \dot{M}/M$, is given via Eq. (2) by $\lambda(t) = \sigma(t) \cdot \phi_R(t)$. From the solution for $\phi_R(t)$, $\lambda(t)$ is predicted to switch from the pre-shift value, $\lambda_i$ for $t < 0$, to the form

$$\lambda(t) = \lambda_0 + \frac{\lambda_f - \lambda_0}{1 + (\lambda_f/\lambda_0)/(e^{\lambda_f t} - 1)}, t \geq 0. \tag{6}$$

The relative increase in mass after a time $t$ from the upshift is obtained by integrating Eq. (6), giving:

$$\frac{M(t)}{M(0)} = 1 + \frac{\lambda_0}{\lambda_f} \left( e^{\lambda_f t} - 1 \right), t \geq 0. \tag{7}$$

**R-proteins abundance determines kinetics in upshifts.** As depicted in Fig. 1b (green line), Eq. (6) describes a transition kinetics featuring an instantaneous jump of the growth rate $\lambda(t)$ from $\lambda_i$ to a larger value $\lambda_0$ at the instant of upshift, followed by a slow adaptation to the final growth rate $\lambda_f$. This jump, whose magnitude is given by the difference $\Delta\lambda \equiv \lambda_0 - \lambda_i$ (Fig. 1b), characterizes the acceleration in transition kinetics from the simple adaptation of $\lambda(t)$ from $\lambda_i$ to $\lambda_f$ according to a logistic equation. As explained in detail in Supplementary Note 1, the jump from $\lambda_i$ to $\lambda_0$ corresponds to a sudden increase in the efficiency of ribosome utilization (from $\sigma_i$ to $\sigma_f$, blue line in Fig. 1b). This rapid increase is a direct reflection of the lack of upstream

bottlenecks in metabolism (e.g., in nutrient uptake and bio-synthesis) assumed in the model, such that the presence of rich medium is immediately made available to the idling ribosomes. The second, slower, phase reflects the progressive accumulation of ribosomes ($\phi_R(t)$, orange line in Fig. 1b, from $\chi_R^i$ to $\chi_R^f$), while keeping the translational efficiency at the post-shift value $\sigma_f$[23, 24]. Within the kinetic model, the value of $\lambda_0$ depends on the pre-shift and post-shift R-sector protein fractions, as well as the final growth rate as

$$\lambda_0 = \chi_R^i \cdot \sigma_f = \phi_R^i \cdot \frac{\lambda_f}{\phi_R^f}. \qquad (8)$$

In particular, Eq. (8) shows that the parameter $\lambda_0$, which captures the acceleration in transition kinetics as explained above, is determined solely by steady state properties of the initial and final states. Eq. (8) can be therefore directly tested by comparing the values of $\lambda_0$ extracted from upshift experiments to the known R-protein abundance as a function of the growth rate.

To test the above predictions, we performed a series of upshifts (Methods section) from defined medium (with pre-shift growth rate $\lambda_i$ ranging between 0.06/h to 0.91/h; Supplementary Table 2) to rich medium (LB + glucose, $\lambda_f = 2.45 \pm 0.2$/h). We show in Fig. 2a, b the growth kinetics for a representative experiments starting from medium with aspartate ($\lambda_i = 0.06 \pm 0.03$/h). Before the nutrient shift, protein mass $M(t)$ (approximately proportional to the optical density of the sample[25]) accumulates exponentially at the pre-shift rates $\lambda_i$. After the shift at time $t = 0$ (dashed line) the rates of mass accumulation increase as the cells adapt to the new growth medium. The growth curves for 9 other shifts are shown in Supplementary Fig. 3. We fitted the observed growth curves (Fig. 2a and Supplementary Fig. 3) to Eq. (7) with $\lambda_0$ being the only fitting parameter. The results are shown as lines, with the values of $\lambda_0$ listed in Supplementary Table 2. The instantaneous growth rate $\lambda(t)$, computed empirically as the discrete time derivative of $\ln M(t)$ for the shifts in Fig. 2a and Supplementary Fig. 3, are shown by the symbols in Fig. 2b and Supplementary Fig. 4, respectively. The data exhibit substantial scatter due to the derivatization; however, the general trends in the data are well captured by the lines generated by Eq. (6), using the values of $\lambda_0$ obtained from the growth curves.

We then test if the experimentally extracted values of $\lambda_0$ for each shift (Supplementary Table 2) follow the theoretical predicted relation (Eq. (8)) with the R-protein content of the cells before the shift, $\phi_R^i$. Indeed, the empirical relation between $\lambda_0$ and $\phi_R^i$ (red circles in Fig. 2c) is in reasonable agreement with the parameter-free prediction (black line), based on Eq. (8); the measured values $\lambda_f = 2.45 \pm 0.2$/h and $\phi_{R,f} = 0.50 \pm 0.01$ were used to generate the black line, with the shaded area reflecting uncertainties in the slope. This agreement confirms that the ribosome abundance in the pre-shift medium is the major factor determining the adaptation kinetics to rich medium, validating a key model assumption that possible effects of metabolic bottle-necks upstream of protein synthesis (e.g., nutrient transport) are negligible.

As a consequence of the growth law in Eq. (1), $\lambda_0$ and $\lambda_i$ are also linearly related, as shown in Fig. 2d; again, the empirical values of $\lambda_0$ (red circles) compare well with the parameter-free prediction indicated by the black line and the shaded area. A slight systematic deviation of the measured $\lambda_0$ from the predicted values in Fig. 2c, d may stem from a slight non-linearity of the measured relation between R-protein and growth rate from the linear growth law presented in Eq. (1) at slow growth rates[10]. Specializing to the case of shift from the very slow to very fast

growth, where $\phi_R^i \approx \phi_{R0}$ and $\phi_R^f \approx \lambda_f / \nu_R$, Eq. (8) becomes

$$\lambda_0 \approx \nu_R \phi_{R0} \equiv \lambda_{R0}, \qquad (9)$$

where we introduced the "R-sector offset" $\lambda_{R0} \equiv \nu_R \phi_{R0}$, which is a rate that gives the magnitude of the x-axis intercept of the R-line given in Eq. (1).

As we will show below, the rate $\lambda_{R0}$, which reflects the magnitude of $\phi_{R0}$, gives the essential time scale for a number of important characteristics involving ribosome overcapacity. For the strain of Escherichia coli we study, $\lambda_{R0}$ is close to 0.43/h (Supplementary Fig. 2). We see that this value is in reasonably good agreement, according to Eq. (9), with the values of $\lambda_0$ measured for shifts from very poor nutrient sources, e.g., $\lambda_0 = 0.52 \pm 0.07$/h for pre-shift growth in Aspartate ($\lambda_i = 0.06 \pm 0.03$/h). Equation (9) has an intuitive explanation: when shifting to rich media, the cell has an immediate boost in protein synthesis ($\Delta\lambda \approx \lambda_0$), whose magnitude is given by the offset $\lambda_{R0}$; this boost is due to the "excess" ribosomes $\phi_{R0}$ being engaged in translation at the maximum efficiency $\nu_R$. Thus, the ribosome overcapacity is quickly activated during these nutritional shifts, effectively acting as a reserve and playing a crucial role in speeding up the transition from famine to feast.

**Fitness landscape for transitions from famine to feast**. For E. coli, remarkably robust empirical laws relate the growth rate and the proteome composition[4, 9, 26]. Such laws can be quantitatively captured by a simple model of proteome allocation[4], relating steady state growth rate and the abundance of the R-sector in terms of only a few parameters (Supplementary Note 2). By jointly using the adaptation kinetics and the protein allocation model, we can predict how strains with different overcapacity $\phi_{R0}$ (and thus different offsets $\lambda_R$) perform when they experience a sudden increase in the nutrient quality from famine to feast.

First of all, the overcapacity $\phi_{R0}$ reduces the maximum possible growth rate during feast. According to the established proteome allocation model[4], the maximum size of the R-sector is limited to a ceiling of $\phi_R^{max} \approx 48\%$ of the proteome. This value is below 100% because of the expression of other non-ribosomal proteins. Because of Eq. (1), growth rate is maximal (e.g., when given the best possible nutrient) when $\phi_R = \phi_R^{max}$, attaining a value $\lambda_{max}$ given by:

$$\lambda_{max} \equiv \nu_R \cdot \left( \phi_R^{max} - \phi_{R0} \right) = \lambda_R^{max} - \lambda_{R0}. \qquad (10)$$

Here $\lambda_R^{max} \equiv \nu_R \phi_R^{max} \approx 2.9$/h is the maximal growth rate possible if there is no R-sector overcapacity (i.e., $\phi_{R0} = 0$). Equation (10) shows that, while the R-sector overcapacity provides a boost to the adaptation kinetics (Eq. (9)), its protein cost reduces the steady state growth rates by reducing $\lambda_{max}$ by an amount given by the offset $\lambda_{R0}$.

Next we consider the situation in which cells with a specific value of $\phi_{R0}$ are shifted from a poor growth medium (barely growing, with $\lambda_i \to 0$) to the best possible post-shift medium ($\lambda_f = \lambda_{max}$), with the latter lasting for a time $T$ (the "feast time") after the shift, as sketched in Fig. 1a. We characterize the growth of these cells by computing the fitness $W(T) \equiv M(T)/M(0)$, defined as the mass increase during the feast time $T$, as given by Eq. (7); see Supplementary Note 2 for analytical expressions of the fitness in terms of parameters of the steady state laws (Eqs. (1) and (10)). We show in Fig. 3a, b the predicted upshift kinetics, for three different values of ribosome overcapacities: The value of our wild-type strain $\phi_{R0}^{WT} = 7\%$ (blue line), one below at $\phi_{R0} = 2\%$ (green line), and one above at $\phi_{R0} = 15\%$ (orange line). The strain with low overcapacity (in green) performs poorly compared to the other two strains if the feast time is short (for $T < 3$ h, Fig. 3b), since it is trapped in the slow recovery phase due to its small jump

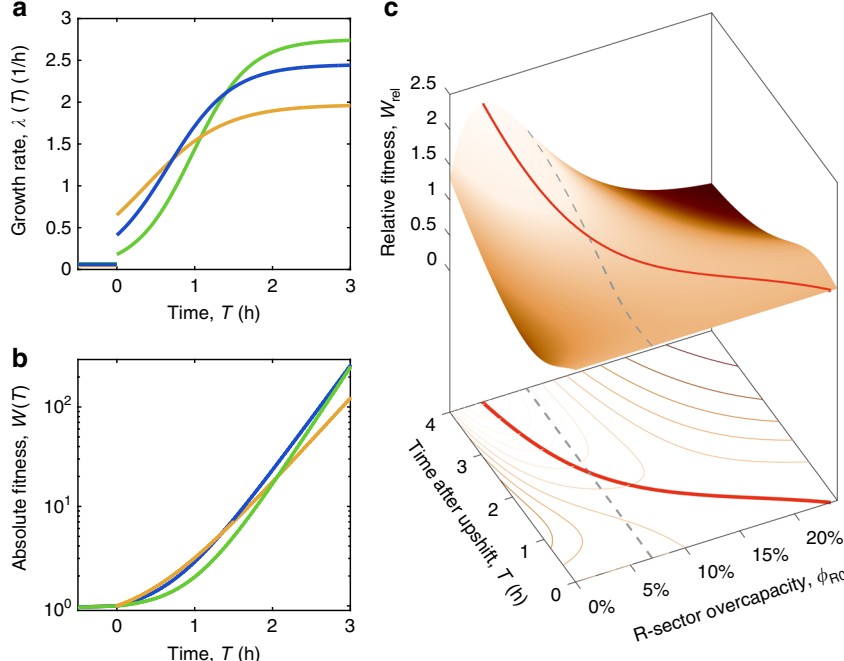

**Fig. 3** Fitness landscape for different protein reserves. **a** Bacterial strains with different overcapacities $\phi_{R0}$ perform differently in upshifts from poor to rich media. Here we show three strains with different overcapacity (2% in green, $\phi_{R0}^{WT} = 7\%$ in blue, 15% in yellow) shifting from a poor nutrient source to a rich media. Strains with large overcapacities grow faster shortly after the upshift to rich media (at $t = 0$), while strains with small overcapacities are advantaged in the long run. **b** Absolute fitness $W(T) = M(T)/M(0)$ (Eq. (7)) obtained by integrating the three instantaneous growth rates $\lambda(T)$ shown in (**a**). **c** Relative fitness landscape for slow ($\lambda_i \to 0$, corresponding to aspartate minimal media) to fast ($\lambda_f \to \lambda_{max}$, corresponding to rich media) growth transitions, as a function of the R-sector proteome overcapacity $\phi_{R0}$ and feast time (time after upshift) $T$. Relative fitness $W_{rel}(\phi_{R0}, T)$ is defined as the absolute fitness $W$, normalized by the average fitness of all other strains considered. Strains with large overcapacities ($\gtrsim 10\%$) are heavily penalized for large feast times ($T \gtrsim 3$ h). Strains with low overcapacities ($\phi_{R0} \lesssim 3\%$, or $\lambda_{R0} \lesssim 0.15$/h) are disadvantaged even many hours after the instant of the upshift, while the ones with large overcapacities are outpaced after a short amount of time. The dashed line indicates the overcapacity of the wild-type strain, while the red line shows the optimal overcapacity $\phi_{R0}^{opt}(T)$ which maximizes $W(\phi_{R0}, T)$ for fixed $T$

in growth rate, $\lambda_0$ (Fig. 3a). Vice versa, the strain with large overcapacity (orange) initially performs well due to the large growth rate jump $\lambda_0$ (Fig. 3a), but it is outperformed by the other strains in the long run (for $T > 1.5$ h, Fig. 3b), since its final growth rate $\lambda_f = \lambda_{max}$ is significantly reduced (Fig. 3a) due to its large ribosome overcapacity as given by Eq. (10). The wild-type strain, with its intermediate value of $\phi_{R0}$, performs well for the range of feast time shown: in fact, the corresponding fitness $W(T)$ (blue line in Fig. 3b) is close to the orange line for short times (<1 h) and to the green line for longer times (around 3 h).

We show in Fig. 3c the "fitness landscape" for the same shift as in Fig. 3a, b, as a function of the feast time $T$ and the overcapacity $\phi_{R0}$. For the ease of display, we show a relative fitness $W_{rel}(T, \phi_{R0}) \equiv W(T, \phi_{R0})/\overline{W}(T)$ obtained by normalizing the fitness $W(T, \phi_{R0})$ by its average $\overline{W}(T)$ across different $\phi_{R0}$ at a given feast time. We also show as a red line the optimal overcapacity $\phi_{R0}^{opt}(T)$, i.e., the overcapacity that maximizes the fitness $W$ for the corresponding feast time $T$. Strains with this overcapacity $\phi_{R0}^{opt}(T)$ are expected to be selected evolutionarily for re-occurring famine-and-feast cycles of feast time $T$. The overcapacity of the wild-type strain is shown as dashed blue line for comparison. At time $T = 0$, all strains have the same fitness, $W_{rel} = 1$. For feast times < 1 h, the best performing strains have large overcapacity, with $\phi_{R0}^{opt} > 10\%$; however, their corresponding fitness values $W_{rel}^{opt}$ are not so significant, i.e., they are not so much better than other values of the overcapacity $\phi_{R0}$, because cellular growth is limited by the short time interval elapsed from the instant of the upshift. As time passes by, strains with smaller overcapacities outcompete the others; the strain with vanishing offset is optimal for steady state growth (recovered in the limit

$T \to \infty$). A very simple relation between $\phi_{R0}^{opt}$ and the feast time $T$ can be derived when $T$ is a few fold larger than $1/\lambda_{max}$ (corresponding approximately to $T > 1$ h):

$$\nu_R \phi_{R0}^{opt} \equiv \lambda_{R0}^{opt} \approx 1/T, \tag{11}$$

i.e, the optimal offset $\lambda_{R0}^{opt}$ is simply given by the reciprocal of the feast time $T$. A comparison of this relation with Eq. (9) shows that the optimal jump in growth rate from poor to rich media ($\lambda_0 \approx \Delta\lambda$) is given by $\lambda_0 \approx \Delta\lambda \approx 1/T$. Based on Eq. (11), our wild-type strain performs optimally in upshifts to rich media characterized by a feast time $T = 1/\lambda_{R0}^{WT}$ that is between 2 and 3 h.

The fitness landscape shown in Fig. 3c is for shifts from very poor growth medium (i.e., famine, characterized by $\lambda_i \to 0$) to very rich medium (i.e., feast, $\lambda_f \to \lambda_{max}$). This analysis can be extended to pre-shift medium supporting generic pre-shift growth rates. To do so, we need to describe the growth rate of strains with arbitrary $\phi_{R0}$ in different medium. We will characterize the quality of the medium by the growth rate it supports for the wild-type strain, denoted as $\lambda^{WT}$. It was shown by Scott et al.[4] that the expression of useless proteins reduced the steady state growth rate in a linear manner, with growth arrest occurring when the useless protein expressed reached $\phi_R^{max}$. Assuming that the ribosome overcapacity $\phi_{R0}$ to exert the same effect on steady state growth rate as a generic useless protein, then we would expect the growth rate $\lambda^*$ to be reduced by $\phi_{R0}$ also in a linear way in the same nutrient condition. With respect to the growth rate of the wild-type strain ($\lambda^{WT}$) with overcapacity $\phi_{R0}^{WT}$,

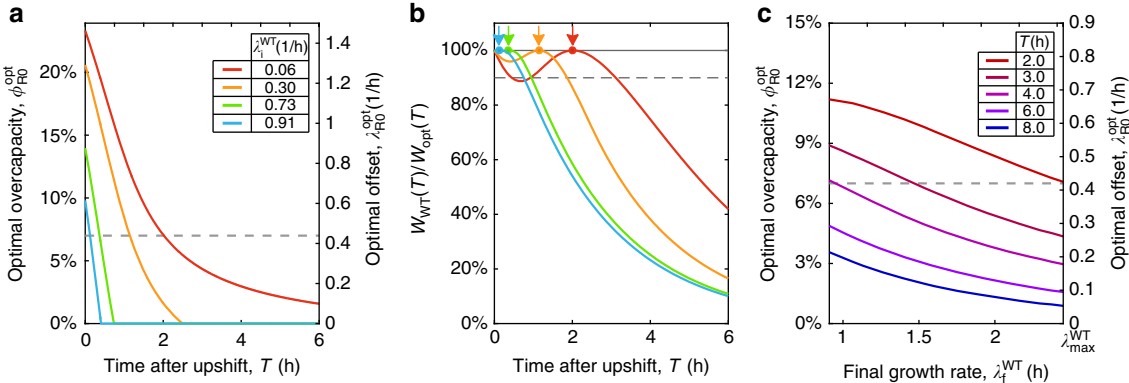

**Fig. 4** Fitness landscape for different upshifts. **a** Optimal overcapacity $\phi_{R0}^{opt}(T)$ as a function of feast time for four different shifts to rich media (from aspartate (red), mannose (orange), glycerol (green), and glucose (blue)), with the pre-shift growth rates for the wild-type strain reported in figure. The overcapacity for the wild-type strain is shown as a dashed line. **b** Fitness of the wild-type strain, $W_{WT}(T) = W(\phi_{R0}^{WT}, T)$, compared to the optimal value $W_{opt}(T) = W(\phi_{R0}^{opt}(T), T)$ for the four shifts shown in (**a**). The wild-type strain is optimal for feast times of 2 h (red arrow), and close to optimal ($W_{WT}/W_{opt} \gtrsim 90\%$) for shifts from poor to rich media (red) for about 3 h. These times are decreased as the pre-shift media improves (orange, green, and blue lines). **c** Optimal overcapacity $\phi_{R0}^{opt}$ as a function of the post-shift growth rate for the wild-type strain, $\lambda_f^{WT}$, varying between 0.91/h (corresponding to the growth rate in glucose minimal medium) and $\lambda_{max}^{WT} = 2.45$/h (rich media). Each line represents a given feast time $T$, ranging between three (red) and 8 h (blue); in all cases the pre-shift growth rate is set to $\lambda_i^{WT} = 0.06$/h (aspartate minimal medium). The dashed line indicates the overcapacity of the wild-type strain

we can write this linear relation as

$$\lambda^*(\phi_{R0}) = \lambda^{WT} \cdot \frac{\phi_R^{max} - \phi_{R0}}{\phi_R^{max} - \phi_{R0}^{WT}}. \qquad (12)$$

This expression allows us to compute the steady state growth rates corresponding to different combinations of nutrient sources (which affect $\lambda^{WT}$) and ribosome overcapacity $\phi_{R0}$.

We show in Fig. 4a the optimal overcapacity $\phi_{R0}^{opt}(T)$ with different feast time $T$, for upshifts from different pre-shift growth medium (characterized by the respective growth rates of the wild type, $\lambda_i^{WT}$) to rich medium; the corresponding growth curves and fitness landscapes are shown in Supplementary Fig. 5. The red line corresponds to shift from a very poor pre-shift medium (aspartate, $\lambda_i^{WT} \approx 0.06$/h) to rich medium, and the dashed line shows the wild-type overcapacity $\phi_{R0}^{WT}$. As the quality of the pre-shift medium improves (in the order of orange, green, and light blue), the value of $\phi_{R0}^{opt}(T)$ decreases more and more rapidly for increasing $T$. The reason is that as pre-shift growth rate increases, the ribosomal content of pre-shift cells also increases in accordance to the growth law Eq. (1). A small initial amount of R-proteins allows the cell to quickly build up the translation machinery (at a rate $\dot{M}_R/M_R = \lambda_f$; Eq. (4)) needed for fast growth, therefore reducing the initial advantage of strains with higher overcapacities.

We show in Fig. 4b the fitness of the wild-type strain $W_{WT}(T) = W(T, \phi_{R0}^{WT})$, as a fraction of the optimal fitness, $W_{opt}(T) = W(T, \phi_{R0}^{opt}(T))$, for the upshifts analyzed in Fig. 4a. We see that the fitness of the wild-type strain in the shift from famine to feast (red line) is maximal for feast times $T \approx 2$ h, as indicated by the red arrow, and does not drop much below 90% of the maximum (dashed line) for all feast times until $T > 3.5$ h. In particular, for small feast times where in principle larger $\phi_{R0}$ would do better (since $\phi_{R0}^{opt}$ increases for smaller $T$ as shown in Fig. 4a), not much advantage is gained in practice over wild-type strain. If the pre-shift medium supports larger growth rates (orange, green, and blue lines), the fitness of the wild-type strain drops quickly for feast time $T$ exceeding 1–2 h.

Our theory can also be extended to describe the case where the quality of the post-shift medium is reduced, i.e., $\lambda_f < \lambda_{max}$, if the instantaneous upshift kinetics assumed here still holds. This

seems to be the case, e.g., for upshift from various poor medium to medium with amino acids but not other ingredients (nucleotides, vitamins, etc) contained in rich medium[16, 21], or if cells growing in carbon-limited chemostat suddenly experience an increase in carbon flux[20]. We see that the optimal overcapacity $\phi_{R0}^{opt}$ increases as $\lambda_f^{WT}$ is decreased from the maximum growth rate of the wild-type strain, $\lambda_{max}^{WT} = 2.45$/h (Fig. 4c). This increase originates in the slower synthesis of R-proteins after the shift, due to the reduced final growth rate: as cells need more time to synthesize the ribosomes, the overcapacity becomes more valuable. Note, however, that for upshift to medium with a simple nutrient source whose uptake requires a large amount of a dedicated transporter, the synthesis of catabolic proteins represents an additional bottleneck, whose understanding requires a more elaborate model. A detailed analysis of the fitness landscape, including a derivation of the expressions involving $\lambda_{R0}^{opt}$, can be found in Supplementary Note 2.

## Discussion

Proteome allocation has been widely studied in the context of cellular economics, both using coarse-grained[4, 6, 27–30] and genome-scale models[31–38]. All of the studies to date have been on steady state systems, and the presence of overcapacities in proteome allocation has been puzzling and counter-intuitive, as they clearly lead to reduced steady state growth[4, 7]. In this work, we characterized the effect of fluctuating environmental conditions on bacterial growth, in an ecological scenario of intermittent growth interrupted by starvation periods, such as the ones experienced by gut microbiota. Using a simple model of upshift kinetics based on the experimental observation for switch to rich medium[22, 24], we elucidated the impact of an overcapacity of the translation machinery (R-proteins) on the kinetics of growth recovery. This overcapacity, which is substantial at low growth rates[10], is suggested to be a reserve that enables rapid growth upon improvement in nutrient conditions.

As a result, we found that cell growth is constrained by a fundamental trade-off, between maximization of steady state exponential growth and rapid growth recovery after nutrient upshift. The optimal solution of the reserve, the one expected to be evolutionarily selected because it maximizes overall growth, would depend strongly on properties of the ecological niche. In

the specific case of famine-to-feast transition studied here, the ecological niche is characterized by the feast time $T$, and the optimal overcapacity depends on $T$ according to Eq. (11). The amount of ribosome reserve maintained by E. coli—different strains have similar overcapacities (Supplementary Fig. 6)—coincides with what the model predicts to be optimal for feasts lasting 2–3 h. Notably, this corresponds to the feast period for the human gut microbiota, since flow rates of nutrient from small to large intestine peak shortly after meals, and decrease considerably after a few hours[39, 40]. This coincidence raises the possibility that this reserve may have been optimized for the typical nutrient cycles in the gut, the major ecological niche where E. coli encounters rich nutrients.

At the regulatory level, synthesis of ribosomal proteins is known to be controlled mainly by the transcription of ribosomal RNA[41] via a tandem pair of promoters[42]. The P2 promoter is regulated by ppGpp to provide demand-dependent expression while the P1 promoter is constitutive[42]. Adjusting the strength of the P1 promoter may therefore be a convenient route for the cell to set the amount of the ribosomal reserve. Interestingly, varying the number of rRNA gene copies in E. coli has been found to modulate both the steady state growth rate and the adaptation kinetics. Systematic, quantitative characterization of the growth kinetics of these strains may be effective ways to test these model predictions. A similar mechanism may underlie the earlier finding that soil bacteria adapting quickly to upshifts have more copies of rRNA genes compared to slowly adapting strains[43]. Adjusting the copy of rRNA genes may thus be another effective way for bacteria to modulate R-sector reserve, thereby allowing them to adjust the adaptation speed with respect to the time scale of environmental changes.

Ribosome reserve is of course not the only type of reserve cells keep for adaptation in fluctuating environments. Metabolically versatile organisms such as E. coli express many genes not needed in a given growth condition. One such class of proteome reserve is comprised of the biosynthetic enzymes, whose availability affect the ability of the cell to adjust its growth following upshifts or downshifts to environments not containing all the metabolic precursors needed for cell growth. Examples include upshift from stationary phase or downshift from rich medium. In these cases, various biosynthetic pathways must be expressed to synthesize the needed precursors; merely keeping a reserve of translational machinery is not sufficient for rapid adaptation. Existing proteomic data shows that most biosynthetic pathways are indeed kept at large overcapacities compared to their metabolic needs in poor nutrient conditions[7]. Quantitative characterization of these shifts will require better description of growth shifts into incomplete medium, which is well beyond the simple kinetics for switch to rich medium used here.

Catabolic proteins comprise another class of protein reserves. For example during growth on a single-carbon substrate the transporters and degradation enzymes of many other sugars are co-expressed with the carbon catabolic system of the carbon source, as seen in recent proteomic studies[7, 44, 45]. Also co-expressed with the carbon catabolic systems is the entire motility system (flagella and motor proteins) which is not needed in, for instance, a well-stirred laboratory culture[7]. The expressions of these proteins have detrimental effects on steady state growth, as shown by directly deleting the flagella system[29], and also in long-term evolution experiments where consistent increases in growth rate are seen for E. coli growing in glucose minimal medium, upon reduction in ribose/maltose catabolism and motility gene expression[46–48]. The fitness cost of expressing these proteins cannot be accounted for by a simple offset, whose effect is a linear reduction of steady state growth rate (as in Eq. (10)), since their expression levels increase upon carbon limitation[6, 7]. The

fitness gain for carrying the catabolic reserves will surely depend on details of the environment, e.g., the amount and duration of the availability of specific types of nutrients[49–51]. This relation underlies a quantitative link between the physiology of the cell and the ecology of the environment, the elucidation of which is one of the outstanding challenges of quantitative systems biology.

## Methods

**Experimental methods.** Growth was performed in a shaking water bath at 250 rpm and 37 °C, using N$^-$ C$^-$ minimal medium[52], containing K$_2$SO$_4$ (1 g), K$_2$HPO$_4$ · 3H$_2$O (17.7 g), KH$_2$PO$_4$ (4.7 g), MgSO$_4$ · 7H$_2$O (0.1 g), and NaCl (2.5 g), supplemented with 20 mM NH$_4$Cl. The pre-shift medium was supplemented with the indicated carbon substrates and E. coli K-12 strain NCM3722[53] grown exponentially as described in ref. [6] until an optical density OD$_{600}$ of about 0.3. Upshift was performed by dilution of the pre-shift culture into fresh, pre-warmed N$^-$ C$^-$ medium containing 2% (w/v) LB, 0.2% glucose and 20 mM NH$_4$Cl to OD$_{600}$ 0.075. Growth was followed until OD$_{600}$ 0.5 and growth rate measured over a 10 min window. The RNA quantification method is based on the method used in ref. [54] with modifications described in ref. [6]. In short, samples were digested in 0.3 M KOH, followed by precipitation of proteins with HClO$_4$. The RNA content in the sample was determined by the absorbance at 260 nm. Total protein was quantified using commercial micro BCA™ assay (Thermo Fisher Scientific Inc., Waltham, MA, USA).

**Data availability**. The data that support the findings of this study are available from the corresponding author on request.

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

## Acknowledgements

We are grateful to members of the Hwa lab for discussion. This work is supported by the NIH (Grant 1R01GM109069) and by the Simons Foundation (Grant 330378) through TH, and by the German Research Foundation via the Excellence Cluster "Nanosystems Initiative Munich" and the priority program SPP1617 (Grant GE1098/6–2) through UG. MM acknowledges the Sapienza Università di Roma for financial support during the early phase of this work. SS acknowledges financial support from the Fulbright Commission.

## Author contributions

All authors designed the study. M.M. and S.S. developed the model and carried out the numerical simulations. S.S. performed the experiments. M.M., S.S., and T.H. wrote the manuscript.

## Additional information

**Competing interests:** The authors declare no competing financial interests.

