## [Peer Review File · Nature Communications]

Reviewers' comments:

Reviewer #1 (Remarks to the Author):

The authors describe a mathematical model of the amount of the growth of cells as a function of the amount of R-sector protein in the cell. They observe that there is an excess amount of this class of proteins in the cell, which hampers achievable steady state growth rates, while allowing the cell to start growing faster.

I am not sufficiently familiar with the immediate related literature (esp ref 9-13). Yet, when seen in a wider context the effect the authors describe may be an instance of a trade-off relationship in a cell. There has been significant work on trade-offs of various types, I believe by the authors themselves as well. It would be interesting to see the relationship between the present work and other trade-offs discussed. For example, it is widely believed that cells can grow fast or adapt fast, but it is not so well understood why this is the case. The present work provides a potential mechanistic explanation for this. It would be very interesting to have this work related more clearly to previous experimental findings.

Another question that remains undiscussed is the relationship between the particular trade-off found here and more fundamental trade-offs between the ability of biological systems to "compute" their own state. There is a large amount of work on such trade-off relationships. I would like to recommend (A. C. Barato and U. Seifert Phys. Rev. Lett. 114, 158101, 2015). This seems unrelated at first, but there could be deep connections between the results reported here and this work.

Another side of this same criticism is that the present MS seems to leave out an important part of the story, namely the speed and cost of regulating the R-sector protein concentration. Imagine that the cell can set the R-sector fraction at any time, instantaneously to any value it "wishes". In this case, there would be no need to worry about maintaining an overcapacity, but the cell could at any time set the ideal ribosome number. The opposite extreme is the case where the fraction is fixed and cannot be changed at all where adaptation is not possible. Over-capacities therefore are related to delays in establishing ideal ss-concentrations. Now, what I think is missing here is a comment on where this delay comes from. Instantaneous control of protein number is clearly not possible, but I would guess that the cell can be tuned to express proteins faster or slower, but this does not come without resource implications. Implicitly, I believe, this is contained in the model, but the reader is somewhat short-changed as the connections are not made very clear.

Part of the cost of accurate control may be stochastic fluctuations of ribosome content. Due to internal noise it may well be that the amount of ribosome expressed by two identical cells may be different due to random variations. One implication of this is that the average number of ribosomes per cell must be much higher than the minimum required in order to ensure that each cell has a reasonable chance of remaining above a minimum number. The overcapacity might just be the consequence of this. (I hasten to add that this does not contradict the authors conclusions, but it would be interesting to learn more about this.)

Credibility of the model: The model is very simple; this is a good thing, but also may make some worry about realism. The authors seems to suggest that the model is well confirmed experimentally. I acknowledge that this may have been discussed in the referenced literature, but given that the entire paper hinges on the model being faithful to reality, it would be good to have some more evidence in the MS. This would also make it more self-contained. The ideal case would be experimental data that corroborates the model. Failing that, it would be good, at least, to provide a more in depth review of the previous literature than is currently provided so as to enable the reader to convince herself.

Minor comments:

-l 78: I believe there should be no dots.

-l 103: Please explain the lag time.

Reviewer #2 (Remarks to the Author):

This study falls within the ongoing projects of T. Hwa on the development of quantitative and empirical bacterial cell models based on resource allocation to explore hypotheses, consequences and potentially some biological experiments to challenge the model.

Here the authors investigate and revisit a classical open biological issue: the apparent overabundance of ribosomes at low growth rates in *E. coli*.

This paper is based on the quite obvious (and shared) idea that a restrictive vision of the allocation of resources in exponential growth phase, i.e. assuming typically that the apparent resource allocation correspond to the optimal allocation in steady-state regime for maximizing growth is contradictory to a rapid adaptation of the growth rate after a nutriment upshift. Clearly, an optimal use of resources (for maximizing growth) in exponential regimen would limit the reserve of power necessary to "speed-up" the transitory phase.

Using a very simple model (which is not a problem by itself, even an advantage), the authors explored the consequences of an overabundance of ribosome at low growth rates. In order to formalize the problem, they translated the overabundance of ribosomes as a loss of "ribosome translation efficiency" at low growth rates. They used a quite standard approach i.e. mass conservation principle and law mass action coupled with ODE, for modeling the simplified cell (even if the authors used normalized mass instead of concentrations). They obtained a quite simple model enabling to compute the expression of various states of the system as an explicit form of time (the computation through concentration leads to the same results). Finally, if the parameters of the model are considered to be constant, closed forms are obtained.

Through this approach, they showed using the closed-form formulae that there exists in the proposed model a trade-off between an optimal use of resources during exponential phase (in particular with respect to the ribosomes and the associated proteins) and the time, or the lag, necessary to reach the new faster exponential regime.

Even if I really appreciate the efforts of the authors to develop and analyze a (simple) mathematical model, their analyses and the paper correspond to my view to a (interesting) preliminary work of investigation of a specific biological question. I agree on the fact that at the stage of cell model development, very simplified models could be really useful. But in this case, the model has to possess some predictive capabilities that enable the design of biological experiments to test/challenge the model and some of its key hypotheses.

Clearly the proposed model and its analysis are quite nice and lead to an attractive explanation about the overabundance of ribosomes. However, to my view, nothing supports this explanation in the proposed paper. Actually, many modelling choices have been made (which is perfectly normal during the modeling process) but these choices have to be precisely discussed, in particular with respect to the existence of alternative explanations. Indeed, there exist alternative models/explanations that are consistent with the speed-up of the cell response to a nutritional upshift. These alternative models have really different premises and led to some extent to different conclusions or at least clearly minor the conclusion of the authors.

The alternative premises are based on the model developed by Marr in 1990 (PMID:1886524). In his model, the author proposed that an important part of the growth rate adaptation of *E. coli* is related to the abundance of the amino-acids in the cell. An alternative major factor of adaptation during a nutritional upshift could be the sudden and almost instantaneous increase of the internal metabolite concentrations (that is in agreement with the recent data). In that context, the ribosome efficiency, through an increase in the concentration of charged tRNAs, could be instantaneously increased as well. The overabundance of ribosome could play a minor role in such a model. In that case, the overabundance of ribosomes at low growth rate that has been suggested in the literature could be for nutritional purposes in case of famine/survival (the role of the Rmf protein / 100S, etc.). This last suggestion has been partially proved experimentally (see e.g. [PMC413246])

Altogether, even if the proposed model and analysis are formally correct and raised interesting explanations, the results to my view are too elusive to be published in Nature.

Reviewer #3 (Remarks to the Author):

Mori et al. aim to explain the seemingly unnecessary expression of many proteins in *E. coli*. Describing the transition from poor to rich media conditions, using few assumptions and a simple proteome allocation model, they show that ribosome overcapacity could provide a fitness advantage by reducing lag times. This gain is traded off with lower growth rates at steady state, especially for the poor conditions. The model predictions for optimal overcapacities correspond well with measured data (only rough estimates are provided). Though the idea that ribosome overcapacity reduces lag times in media shifts has been around for a while, a quantitative description of this phenomenon is novel. Furthermore, the paper clearly states the assumptions made and its conclusions are mathematically valid.

Although this paper focuses on the theory, many of the conclusions can be tested experimentally. The authors allude to a few such experiments, but mostly to qualitative results. An experimental validation of the qualitative results (i.e. manipulating the overcapacity and measuring fitness advantages for different post-shift growth rate ratios and pulse times) would be very useful. Nevertheless, the theoretical results and conclusions of this work stand on their own.

One major concern is the writing style of the manuscript, which uses mathematical language extensively. Most of it is essential for the message of the work, but the authors can do more to help readers who might struggle with it. For instance, it would be quite useful to have a table listing all the variables defined in the manuscript and their descriptions - there are about 8 different letters, most having 2-3 subscripts. Specifically, the lag time τ_{lag} is not defined in the main text and a reference to the SI is missing. Also, more functions could be visualized with plots.

In conclusion, Mori et al. manage to find an elegant solution to a question that has bothered microbiologists for a long time. However, the significance of this work could be improved if more experimental evidence was provided and if the text was made more accessible.

REVIEWER 1

The authors describe a mathematical model of the amount of the growth of cells as a function of the amount of R-section protein in the cell. They observe that there is an excess amount of this class of proteins in the cell, which hampers achievable steady state growth rates, while allowing the cell to start growing faster.

I am not sufficiently familiar with the immediate related literature (esp ref 9-13). Yet, when seen in a wider context the effect the authors describe may be an instance of a trade-off relationship in a cell. There has been significant work on trade-offs of various types, I believe by the authors themselves as well. It would be interesting to see the relationship between the present work and other trade-offs discussed. For example, it is widely believed that cells can grow fast or adapt fast, but it is not so well understood why this is the case. The present work provides a potential mechanistic explanation for this. It would be very interesting to have this work related more clearly to previous experimental findings.

Trade-offs are extensively studied throughout many fields of biology, and in the field of bacterial growth many of such trade-offs can be traced to their mechanistic origins. The Reviewer might be referring to a recent example from our group which studies the trade-off between different modes of metabolism (fermentation versus respiration) under the constraint of maximizing the ribosomal output (Basan et al. 2016, Ref. [28]). In the submitted work, we extend this style of analysis to a trade-off between growth and adaptation in the ecological context. As the Reviewer noted, “it is widely believed that cells can [either] grow fast or adapt fast”. We are convinced that the overcapacity of ribosomes (several thousands are kept even in stationary phase!) is crucial to speed up the recovery process after nutrient resupply. At the same time, we can trace the cost of the overcapacity to its mechanistic origins (they occupy a significant share of the protein synthesis machinery). Since we can understand both cost and benefit quantitatively, this allows us to examine the trade-off these quantities in detail.

We do not want to dismiss the role of other trade-offs between fast growth or quick adaptation. Below, the Reviewer mentioned stochasticity and the cost of accurate regulation, to which we will respond in detail. Metabolic trade-offs, such as studied by Basan et al. [28] are likely to play a role if the nutrient shift involves shifts in competing biosynthetic pathways. In this work such additional tradeoffs are avoided by focusing on shifts to rich media, which are replete with most metabolic precursors. This allows us to isolate and characterize the effect of ribosome allocation on growth kinetics.

We kindly ask the Reviewer to provide examples of other trade-offs that he/she would like to see discussed in the context of this paper.

Another question that remains undiscussed is the relationship between the particular trade-off found here and more fundamental trade-offs between the ability of biological systems to “compute” their own state. There is a large amount of work on such trade-off relationships. I would like to recommend (A. C. Barato and U. Seifert Phys. Rev. Lett. 114, 158101, 2015). This seems unrelated at first, but there could be deep connections between the results reported here and this work.

The reviewer is suggesting a highly interesting work, which identifies the cost of thermodynamic uncertainty in many biological processes. This approach can be applied to processes that can be described by a Markov process, e.g. 1D processes like molecular motors, but also more complex networks. Applied to ribosome synthesis, the uncertainty as defined by the thermodynamic uncertainty relation is the uncertainty in the time to translate a single protein. According to Barato and Seifert, maintaining a 10% uncertainty in the timing would cost $200k_B T$, a negligible value compared to the $\sim 40,000$ ATP (corresponding to $\sim 800,000 k_B T$) and 7,500 amino acids consumed during synthesis of a single ribosome.

Similarly, the cost of regulating gene expression is low compared to the cost of protein synthesis: regulatory proteins, RNAs and signaling molecules can sufficiently control gene expression at low mass fraction. For example the expression of the Lac operon ($\sim 1\%$ of the proteome mass when induced) is regulated by as few as 10 tetrameric copies of the lac repressor LacI ($< 0.01\%$ of the mass fraction).

Stochasticity, and its reduction/control, thus plays only a minor role of the cost of gene expression. The dominant cost

of protein expression originates from occupation of the translational machinery, which was quantified and understood using a proteome allocation model: For *E. coli*, when 1% of the proteome is allocated to useless protein (in our case an overcapacity of ribosomes), growth rate is reduced by about 2%, as experimentally established in Ref. [4].

Another side of this same criticism is that the present MS seems to leave out an important part of the story, namely the speed and cost of regulating the R-sector protein concentration. Imagine that the cell can set the R-sector fraction at any time, instantaneously to any value it “wishes”. In this case, there would be no need to worry about maintaining an overcapacity, but the cell could at any time set the ideal ribosome number. The opposite extreme is the case where the fraction is fixed and cannot be changed at all where adaptation is not possible. Over-capacities therefore are related to delays in establishing ideal ss-concentrations. Now, what I think is missing here is a comment on where this delay comes from. Instantaneous control of protein number is clearly not possible, but I would guess that the cell can be tuned to express proteins faster or slower, but this does not come without resource implications. Implicitly, I believe, this is contained in the model, but the reader is somewhat short-changed as the connections are not made very clear.

As the reviewer correctly observes, R-proteins cannot be “tuned” instantaneously to any desired level, because they need to be synthesized by the cell. Their mass fraction is substantial: R-proteins range between 10% of the proteome at slow growth to up to 50% at the fastest growth conditions (Ref. [4]). The only way to increase the R-protein fraction in a shift from slow to fast growth is to increase its R-protein synthesis \dot{M}_R relative to the synthesis of other proteins. This is explicitly encoded in our theory in the new Eq. (4): $\dot{M}_R = \chi_R \dot{M}$, where χ_R is the fraction of total protein synthesis \dot{M} directed to R-proteins. This results in a logistic equation for the R-protein mass fraction with the time scale (i.e. delay) given by the inverse growth rate, see new Eq. (5). The resulting kinetics for growth rate and biomass Eqs. (6-7) are directly validated in the new Fig. 2.

To help the Reviewer and other readers grasp our model, we have extended the exposition of the kinetic model, and we now explicitly discuss the logistic function and associated time scale (i.e. delay) around Eq. (5).

Part of the cost of accurate control may be stochastic fluctuations of ribosome content. Due to internal noise it may well be that the amount of ribosome expressed by two identical cells may be different due to random variations. One implication of this is that the average number of ribosomes per cell must be much higher than the minimum required in order to ensure that each cell has a reasonable chance of remaining above a minimum number. The overcapacity might just be the consequence of this. (I hasten to add that this does not contradict the authors conclusions, but it would be interesting to learn more about this.)

The Reviewer is raising the point of stochastic gene expression of ribosomes and their potential to limit growth. Such stochastic limitations are observed when growth is limited by the expression of a highly stochastic protein, often due to low copy numbers. Some examples can be found in Megerle et al. (2008) (doi: 10.1529/biophysj.107.127191) and Kiviet et al. (2014) (doi:10.1038/nature13582). Ribosomes, however, are expressed in excess, such that at slow growth 80% of ribosomes are inactivated and stored by various mechanisms. Inactivation was recently understood as a way to maintain high elongation rates of the few remaining active ribosomes (Ref. [36]). The drastic overexpression, on the other hand, remains puzzling. Due to the high cost of overexpression (7% reserve lead to 15% growth reduction) and the complex inactivation mechanisms, we believe that it is highly unlikely that this overexpression is simply the result of an incapability of the cell to properly regulate itself.

Credibility of the model: The model is very simple; this is a good thing, but also may make some worry about realism. The authors seems to suggest that the model is well confirmed experimentally. I acknowledge that this may have been discussed in the referenced literature, but given that the entire paper hinges on the model being faithful to reality, it would be good to have some more evidence in the MS. This would also make it more self-contained. The ideal case would be experimental data that corroborates the model. Failing that, it would be good, at least, to provide a more in depth review of the previous literature than is currently provided so as to enable the reader to convince herself.

We thank the Reviewer for appreciating our effort on working with a minimal model. The fact that we do not directly confirm our theory experimentally, was indeed a weak point of our study, criticized by all reviewers. In the revised version of the manuscript we now validate our theory directly. We hope that the parameter free match of the growth theory with upshift experiment (Fig. 2) suffices to convince the Reviewer of the realism of this model.

REVIEWER 2

This study falls within the ongoing projects of T. Hwa on the development of quantitative and empirical bacterial cell models based on resource allocation to explore hypotheses, consequences and potentially some biological experiments to challenge the model.

Here the authors investigate and revisit a classical open biological issue: the apparent overabundance of ribosomes at low growth rates in *E. coli*.

This paper is based on the quite obvious (and shared) idea that a restrictive vision of the allocation of resources in exponential growth phase, i.e. assuming typically that the apparent resource allocation correspond to the optimal allocation in steady-state regime for maximizing growth is contradictory to a rapid adaptation of the growth rate after a nutriment upshift. Clearly, an optimal use of resources (for maximizing growth) in exponential regimen would limit the reserve of power necessary to “speed-up” the transitory phase.

Using a very simple model (which is not a problem by itself, even an advantage), the authors explored the consequences of an overabundance of ribosome at low growth rates. In order to formalize the problem, they translated the overabundance of ribosomes as a loss of “ribosome translation efficiency” at low growth rates. They used a quite standard approach i.e. mass conservation principle and law mass action coupled with ODE, for modeling the simplified cell (even if the authors used normalized mass instead of concentrations). They obtained a quite simple model enabling to compute the expression of various states of the system as an explicit form of time (the computation through concentration leads to the same results). Finally, if the parameters of the model are considered to be constant, closed forms are obtained.

We thank the reviewer for mentioning this point, which we neglected to mention in the previous version of the manuscript. Protein mass fractions (used in this work) are proportional to protein concentrations, because the “total protein density” ρ_{TP} (computed as total protein mass per cell, over the cell volume) is relatively constant for different growth rates, as shown in the new Fig. S1, computed using data from Basan et al. (2015), Ref. [25]. As a consequence, the concentration of each protein is directly proportional to its mass fraction through a constant (growth-independent) factor. We now mention this equivalence in the introduction (lines 28-30) and reference the new Fig. S1, when normalized masses are first introduced.

Through this approach, they showed using the closed-form formulae that there exists in the proposed model a trade-off between an optimal use of resources during exponential phase (in particular with respect to the ribosomes and the associated proteins) and the time, or the lag, necessary to reach the new faster exponential regime.

Even if I really appreciate the efforts of the authors to develop and analyze a (simple) mathematical model, their analyses and the paper correspond to my view to a (interesting) preliminary work of investigation of a specific biological question. I agree on the fact that at the stage of cell model development, very simplified models could be really useful. But in this case, the model has to possess some predictive capabilities that enable the design of biological experiments to test/challenge the model and some of its key hypotheses.

Clearly the proposed model and its analysis are quite nice and lead to an attractive explanation about the overabundance of ribosomes. However, to my view, nothing supports this explanation in the proposed paper. Actually, many modelling choices have been made (which is perfectly normal during the modeling process) but these choices have to be precisely discussed, in particular with respect to the existence of alternative explanations.

We thank the Reviewer for appreciating our efforts and for his/her general appreciation of the usefulness of minimal models, such as the one used in our work here. In response, to the Reviewer’s demand that our model needs to “possess [...] predictive capabilities”, the revised version of the manuscript now includes experimental validation of our theoretical model. In Fig. 2 and Fig. S3/S4 we tested our growth shifts model by measuring growth shift kinetics. Growth follows the predicted kinetics, and the initial boost of growth rate λ_0 can be quantitatively traced back to the ribosome overcapacity. We hope that this experimental validation convinces the Reviewer to believe in the realism of our modeling approach.

Indeed, there exist alternative models/explanations that are consistent with the speed-up of the cell response to a nutritional up-shift. These alternative models have really different premises and led to some extent to different

conclusions or at least clearly minor the conclusion of the authors.

The alternative premises are based on the model developed by Marr in 1990 (PMID:1886524). In his model, the author proposed that an important part of the growth rate adaptation of *E. coli* is related to the abundance of the amino-acids in the cell. An alternative major factor of adaptation during a nutritional upshift could be the sudden and almost instantaneous increase of the internal metabolite concentrations (that is in agreement with the recent data). In that context, the ribosome efficiency, through an increase in the concentration of charged tRNAs, could be instantaneously increased as well. The overabundance of ribosome could play a minor role in such a model.

The question raised by the reviewer is an important one. The internal concentration of amino acids (and other precursors) certainly affects protein synthesis, e.g. by modulating the abundance of charged tRNAs and, hence, the elongation rate. However, an increase in amino acid abundance alone cannot account for the increased protein synthesis rate. Koch, Brunschede and others directly measured the average protein synthesis rate/ ribosome (called translational capacity σ in this work) and observed an eruption (increase of several fold) immediately after up-shift (Koch & Deppe (1971), ref. [19]; Koch (1971), ref. [14], Brunschede *et al.* (1977), ref. [15]). However, in a recent work by Dai *et al.* (Ref. [36]), it was shown that in very slowly growing cells the elongation rate (or ribosome efficiency as called by the Reviewer) is only two fold smaller than in fast growing cells. Therefore, an increase of elongation rate by higher amino acid abundance cannot account for the several fold increase of protein synthesis observed by Koch, Brunschede and others.

To reconcile the small protein synthesis flux and the (relatively) large elongation rate, Dai *et al.* argue that at slow growth the majority of ribosomes have to remain somehow idle or non-functional. The authors estimated that the active (non-idle) ribosome fraction at extremely slow growth is below 20% of that in fast growth (their Fig. 3C). The immediate activation of the remaining 80% of the ribosomes, plus the smaller increase in the elongation rate due to amino acid levels, can therefore account for the long-standing puzzle posed by the observations of Brunschede *et al.* and Koch *et al.* for these nutrient upshifts. In our model, both the elongation rate (k) and the fraction of active ribosomes (f_{act}) are effectively accounted for in the translational capacity, which can be expressed as the product of the two as $\sigma = k_{\text{el}} \cdot f_{\text{act}}$. We now mention this in the main text (lines 73-74) and included a discussion of the origin of the translation efficiency in Supplementary Note 1.1.

We argue that our modeling choices are deeply rooted in well-characterized and consolidated phenomenological laws, i.e. the abundance of R-proteins as a function of (steady state) growth rate (see Ref. [4]), and the sudden increase in protein and ribosomal proteins/stable RNA synthesis for nutritional shifts to rich media (see Refs. [11, 21, 22]). This allows us to provide a faithful description of the upshift kinetics without referring to any underlying microscopic model, which would be necessarily more complex; vice versa, microscopic models such as the one indicated by the reviewer (Marr (1991)) would provide a mechanistic description of the parameters entering in our framework (most notably, the translation efficiency σ and the R-sector synthesis fraction χ_R). This would however come at the cost of and increased complexity of the model itself, without any guarantee of obtaining a better agreement with the data: for instance, the model in Marr (1991) does not quantitatively describe upshifts kinetics (Fig. 15 in Marr's paper reveals a poor agreement between model and data; compare with Fig. 2b in our manuscript).

In that case, the overabundance of ribosomes at low growth rate that has been suggested in the literature could be for nutritional purposes in case of famine/survival (the role of the Rmf protein / 100S, etc.). This last suggestion has been partially proved experimentally (see e.g. [PMC413246])

Dai *et al.* (Ref. [36]) suggested that a reduction of translating ribosomes (possibly via Rmf/100S) enables *E. coli* to maintain elongation rates even at slow growth. It sounds plausible that these inactivated ribosomes are consumed for their nutrients if cells enter famine/survival, as observed by Mandelstam and Halvorson (1960) ([https://doi.org/10.1016/0006-3002\(60\)91313-5](https://doi.org/10.1016/0006-3002(60)91313-5)) or Yamagishi *et al.* [PMC413246]. However, ribosomes being only built to be inactivated and stored as a nutrient source sounds overly complicated, given other mechanisms that could provide nutrients storage, e.g. glycogen.

Altogether, even if the proposed model and analysis are formally correct and raised interesting explanations, the results to my view are too elusive to be published in Nature.

We hope that having put our theory on a solid experimental basis will convince the Reviewer that the quantitative

trade-off established theoretically in this work is concrete rather than elusive in the revised manuscript and that it deserves publication in Nature Communications.

REVIEWER 3

Mori et al. aim to explain the seemingly unnecessary expression of many proteins in *E. coli*. Describing the transition from poor to rich media conditions, using few assumptions and a simple proteome allocation model, they show that ribosome overcapacity could provide a fitness advantage by reducing lag times. This gain is traded off with lower growth rates at steady state, especially for the poor conditions. The model predictions for optimal overcapacities correspond well with measured data (only rough estimates are provided). Though the idea that ribosome overcapacity reduces lag times in media shifts has been around for a while, a quantitative description of this phenomenon is novel. Furthermore, the paper clearly states the assumptions made and its conclusions are mathematically valid.

Although this paper focuses on the theory, many of the conclusions can be tested experimentally. The authors allude to a few such experiments, but mostly to qualitative results. An experimental validation of the qualitative results (i.e. manipulating the overcapacity and measuring fitness advantages for different post-shift growth rate ratios and pulse times) would be very useful. Nevertheless, the theoretical results and conclusions of this work stand on their own.

We thank the Reviewer for appreciating the novelty of the quantitative trade-off found in this paper. This Reviewer notes that our “conclusions can be tested experimentally”. We agree with the reviewer and in the revised version of the manuscript we put our theoretical consideration on a solid experimental basis. In Fig. 2 and Fig. S3/S4 we validated our growth shifts model by measuring growth shift kinetics for ten upshifts from slow growth (0.06/h to 1/h) to fast growth (2.45/h). Growth follows the predicted kinetics, and the initial boost of growth rate λ_0 can be traced back to the ribosome overcapacity. Currently, however, there is no way to directly manipulate the overcapacity, due to the complicated regulatory mechanisms governing ribosome expression.

One major concern is the writing style of the manuscript, which uses mathematical language extensively. Most of it is essential for the message of the work, but the authors can do more to help readers who might struggle with it. For instance, it would be quite useful to have a table listing all the variables defined in the manuscript and their descriptions - there are about 8 different letters, most having 2-3 subscripts. Specifically, the lag time τ_{lag} is not defined in the main text and a reference to the SI is missing. Also, more functions could be visualized with plots.

As suggested by the reviewer, in the revised manuscript we provide a list of all variables in Table S1; and we show in Fig. 1b the time-dependence of the model's parameters in a shift from poor to rich media. We further carefully defined all variables (and sub- and superscripts) in the main text; lag time τ_{lag} was not essential for carrying the message of the paper and was thus removed from the main text to reduce the mathematical complexity. We hope that these efforts help readers from diverse backgrounds to better grasp our assumptions and conclusions.

In conclusion, Mori et al. manage to find an elegant solution to a question that has bothered microbiologists for a long time. However, the significance of this work could be improved if more experimental evidence was provided and if the text was made more accessible.

Reviewers' comments:

Reviewer #1 (Remarks to the Author):

I am satisfied with the changes and recommend this MS for publication.

Reviewer #2 (Remarks to the Author):

I have to confess that I did not read for the previous review a (very interesting) reference of the last author around the translation speed of E.coli (ref. Dai et al 2016). In this reference, it is shown that even at a very low growth rate, the speed of translation remains high (around 7 aa/s). This simple observation implies that a large part of ribosome is not active at slow growth rates. As it is discussed in Dai et al 2016, this quite interesting observation leads to an immediate and interesting question around the fact that E. coli has at low growth rates a large excess of ribosome.

Obviously, an immediate issue is to explain why such an excess does exist? Its interest? This question is especially interesting in the general context of the growth rate maximization and the fact that the accumulation of "unnecessary" proteins (&RNA) leads in principle to decrease the growth rate of a bacterium... So, theoretically, a bacterium is able by decreasing its excess of ribosome to increase its growth rate at least at low growth rate.

Following this general context, the goal of the first part of the paper is to show that the excess of ribosome at low growth rates allow the bacteria to speed-up the transitory phase of an upshift from a poor to a rich medium. In particular, as it already known, there exists a jump of the growth rate just after the upshift. The hypothesis of the authors is that the jump and more generally the behavior of the growth rate is a consequence of the fact that immediately after the addition of nutrients:

(1) all the available ribosomes are now active and the translation speed is now the one associated to the rich medium;

(2) The ribosomes (actually all the components associated to the translation) are now produced at a rate corresponding to the one associated to the steady state of the rich medium.

On this basis of these assumptions, the authors develop a simple (elegant) empirical model (based on previous works of the last authors) and in view of the chosen assumptions, they obtain a closed-form formula of the time behavior of the growth rate. This formula allows the authors to compute from the experimental data, the value of the jump (and the growth rate transitory phase).

Before the paper of Dai et al 2016, the status of the translation speed at (very) low growth rate is unclear (at least for the reviewer) but the fact that ribosome at low growth rate be in excess is generally accepted. The existence of a jump on the growth rate (and the growth rate transitory phase) is explained in this context by an immediate reprogramming of the metabolic network and the existence of jumps on the concentration of some internal metabolites (especially amino-acids). These jumps on the internal metabolite concentrations lead to increase the translation speed and thus to produce a jump on the growth rate....

Clearly, the recent contribution presented in Dai et al 2016 leads to a finer picture of the problem since it is established that a large fraction of ribosome is not active at low growth rates. Hence, the jump (and more generally the behavior of the growth rate with respect to time) could be then due to a more sophisticated mechanism combining a metabolic adaptation and the recruitment of non-active ribosomes.

By the way, the main assumption of the authors is to say that the translation is the main actor of the adaptation. By definition, in view of the link between the translation and the growth rate, the translation is by definition a key actor. However, the previous claim is true if an implicit assumption is made: the metabolic part of bacteria is able to provide the energy and all the elements necessary for producing such huge quantity of protein and more generally is able to

support the growth of the bacteria by furnishing the bacterial components as DNA, membrane, cell wall, etc.

So a first and immediate corollary of the presented analysis is clearly the fact that the enzyme concentrations associated to the metabolic part have to be enough abundant in order to be able to furnish the energy and all the biomass components in accordance to the increasing growth rate ...leading a first unsolved question of how it is possible for the metabolic part to achieve such a goal? (as for ribosome, with an excess of enzymes for example?)

A second corollary is a consequence of the strong relation between the three main fractions of protein in the bacteria. Indeed, when the authors assume that the proportion of ribosomal protein changes, that means through the rules existing between the different fractions (assumed in Scott et al) that for example the fraction associated to the metabolic part has to change in accordance...(decrease).

Obviously, I understand the fact that it is necessary and useful to identify simple rules explaining bacteria behavior. However, it remains necessary to clarify the implicit assumption and its consequences on the result.

So three main points has to be addressed

1 / It is essential to emphasize more clearly the hypotheses concerning the metabolism and the constraint relied to the evolution of the protein fraction dedicated to the translation. Indeed, that implies at least the existence of a coordination ensuring the consistence of this constraint. Finally, that allows another time to insist on the remarkable simplicity of the growth rate response with regard to the complexity of a bacterium.

2/ The issue related to the metabolism leads to an important question. Indeed, it is known that some enzymes necessary at a low growth rate are no longer necessary in the rich medium. So if we suppose that they are, as the ribosome, also in excess, at low growth rate, this raises the question of their uselessness and their disappearance by 'dilution' (and the fact that they occupy for a while a place in the cytosol).

3 / Finally, I do not understand why the authors do not position more clearly this work with respect to Dai et al. 2016. Indeed, it constitutes an important premise of this work. Moreover, with regard to the data produced in Dais et al 2016, it seems possible to compare the values of λ_0 obtained experimentally with the value that can be deduced with the data of Bai et al. 2016. In particular, in view of Bai et al., it seems also possible to refine the parameters associated to the model of Scott et al?

The second part of the paper is from my point of view speculative. The presented arguments and the associated computations do not provide more than expected. The question here is not to say that this part is incorrect, only to say that it provides a rational leading to think that the maximization of the growth rate is not necessarily the best solution with respect to specific ecologic niches...

It seems to my view more interesting to point out that the bacteria has to manage the resources with respect to the Scott et al. and by taking into account another dimension of the problem: the necessity to use a part of resources to manage transitory/dynamical phase.... This part of the article then points out to my understanding that this new trade-off is (obviously) strongly dependent on the ecologic niche and the considered scenario ...

Reviewer #3 (Remarks to the Author):

The authors have carefully addressed the previous concerns and the manuscript is greatly improved. The experimental validation of the model is an important addition and the derivations are much easier to follow. Regarding the experimental validation, I separate it into two parts. For

the steady-state data presented in figure S2 and S6, there seems to be enough data to establish a linear relationship between growth rate and R/P mass ratio. However, the assumption of instantaneous shift of σ and χ is based on previous studies. Nevertheless, how well the model presented in this paper fits the measurements can be seen as a validation of this assumption.

I feel that this analysis is compelling, but some statistical aspects should be improved in order to have a strong case for the model:

- In figure 2b, the authors claim there is substantial scatter due to the time derivative. A proper estimation of the error would be very helpful in this case (visualized as error bars in the plot). Same goes also to figure S4, where the fits are even poorer.
- The model predicts λ_0 as a function of λ_i (for the experimental setup where λ_f was kept constant). Why does figure 2c show the best fit line and not the model prediction? Instead, figure 2d shows λ_0 as a function of ϕ_{Ri} together with the model prediction. This is a peculiar decision, especially given that ϕ_{Ri} is simply a linear transformation of λ_i using the parameters from figure S2, so that would yield $\lambda_0 = 0.36 + 0.82 * \lambda_i$. I fail to see the point of showing the best fit in Fig2c and its parameters if they are not used anywhere else. Also, if we were to believe the provided fit, that would mean that there would be a zero jump in growth rate when $\lambda_i = 0.46/(1-0.59) = 1.12/h$. This is a very different value than the expected 2.45/h.
- In Figure S3, model fitting was done in linear scale, which is not a natural scale for fitting an exponential function. In this case, a semilog-y plot would not make it linear, but could still help guide the eye. Alternatively, the authors could consider transforming the x-axis using $t \rightarrow \exp(\lambda_f * t) - 1$, which should yield a linear relationship if the model is correct.
- From looking at figure S4, it looks like there should be large uncertainty in the value of λ_0 and that should also be reflected in figure 2c-d as error bars.

Minor comments:

Line 169: "overcapacitie" → overcapacity

Line 187: "defined as as" → remove one 'as'

Line 206: "at a givern feast time" → given

Line 219: "jump in growth rate at the from poor to rich" → remove 'at the'

Line 230: "effect on stead state" → 'steady state'

Reviewer #1

I am satisfied with the changes and recommend this MS for publication.

We thank the Reviewer for his/her support!

Reviewer #2

I have to confess that I did not read for the previous review a (very interesting) reference of the last author around the translation speed of *E. coli* (ref. Dai et al 2016). In this reference, it is shown that even at a very low growth rate, the speed of translation remains high (around 7 aa/s). This simple observation implies that a large part of ribosome is not active at slow growth rates. As it is discussed in Dai et al 2016, this quite interesting observation leads to an immediate and interesting question around the fact that *E. coli* has at low growth rates a large excess of ribosome.

Obviously, an immediate issue is to explain why such an excess does exist? Its interest? This question is especially interesting in the general context of the growth rate maximization and the fact that the accumulation of “unnecessary” proteins (&RNA) leads in principle to decrease the growth rate of a bacterium... So, theoretically, a bacterium is able by decreasing its excess of ribosome to increase its growth rate at least at low growth rate.

Following this general context, the goal of the first part of the paper is to show that the excess of ribosome at low growth rates allow the bacteria to speed-up the transitory phase of an upshift from a poor to a rich medium. In particular, as it is already known, there exists a jump of the growth rate just after the upshift. The hypothesis of the authors is that the jump and more generally the behavior of the growth rate is a consequence of the fact that immediately after the addition of nutrients:

- (1) all the available ribosomes are now active and the translation speed is now the one associated to the rich medium;
- (2) The ribosomes (actually all the components associated to the translation) are now produced at a rate corresponding to the one associated to the steady state of the rich medium.

On this basis of these assumptions, the authors develop a simple (elegant) empirical model (based on previous works of the last authors) and in view of the chosen assumptions, they obtain a closed-form formula of the time behavior of the growth rate. This formula allows the authors to compute from the experimental data, the value of the jump (and the growth rate transitory phase).

Before the paper of Dai et al 2016, the status of the translation speed at (very) low growth rate is unclear (at least for the reviewer) but the fact that ribosome at low growth rate be in excess is generally accepted. The existence of a jump on the growth rate (and the growth rate transitory phase) is explained in this context by an immediate reprogramming of the metabolic network and the existence of jumps on the concentration of some internal metabolites (especially amino-acids). These jumps on the internal metabolite concentrations lead to increase the translation speed and thus to produce a jump on the growth rate....

Clearly, the recent contribution presented in Dai et al 2016 leads to a finer picture of the problem since it is established that a large fraction of ribosome is not active at low growth rates. Hence, the jump (and more generally the behavior of the growth rate with respect to time) could be then due to a more sophisticated mechanism combining a metabolic adaptation and the recruitment of non-active ribosomes.

By the way, the main assumption of the authors is to say that the translation is the main actor of the adaptation. By definition, in view of the link between the translation and the growth rate, the translation is by definition a key actor. However, the previous claim is true if an implicit assumption is made: the metabolic part of bacteria is able to provide the energy and all the elements necessary for producing such huge quantity of protein and more generally is able to support the growth of the bacteria by furnishing the bacterial components as DNA, membrane, cell wall, etc. So a first and immediate corollary of the presented analysis is clearly the fact that the enzyme concentrations associated to the metabolic part have to be enough abundant in order to be able to furnish the energy and all the biomass components in accordance to the increasing growth rate ...leading a first unsolved question of how it is possible for the metabolic part to achieve such a goal? (as for ribosome, with an excess of enzymes for example?)

A second corollary is a consequence of the strong relation between the three main fractions of protein in the bacteria. Indeed, when the authors assume that the proportion of ribosomal protein changes, that means through the rules existing between the different fractions (assumed in Scott et al) that for example the fraction associated to the metabolic part has to change in accordance... (decrease). Obviously, I understand the fact that it is necessary and useful to identify simple rules explaining bacteria behavior. However, it remains necessary to clarify the implicit assumption and its consequences on the result.

We thank the reviewer for the efforts to understand our work in the context of the existing literature. We understand that we need to state our assumptions more clearly, and we updated the manuscript accordingly. We answer the three main points raised by the reviewer below.

So three main points has to be addressed

1 / It is essential to emphasize more clearly the hypotheses concerning the metabolism and the constraint relied to the evolution of the protein fraction dedicated to the translation. Indeed, that implies at least the existence of a coordination ensuring the consistence of this constraint. Finally, that allows another time to insist on the remarkable simplicity of the growth rate response with regard to the complexity of a bacterium.

The Reviewer asked us to state more clearly the assumptions made on metabolism, i.e. how it can keep up with the increased demand for building blocks. To answer this point, we want to emphasize that by design, our work focuses on nutrient upshifts to *rich* medium, which contains all the essential building blocks such as amino acids and nucleotides, and also membrane components. This alleviates the need for the cells to synthesize these essential building blocks in the post-shift phase. In the revised manuscript, we added a statement on the usage of rich medium, see lines 72 to 78, for which the limitation of cell growth by ribosomes and the affiliated machinery is a reasonable approximation. We further provided two references which found the average protein synthesis rate to increase several folds after upshift to rich medium, thus strongly supporting our assumptions.

Second, the reviewer asked us to emphasize the constraint on the time evolution of the protein abundances. As the reviewer correctly states, there exists a coordination of the proteome fractions, in particular between the ribosomal synthesis fraction χ_R and the synthesis fraction of non-ribosomal proteins. If the ribosomal synthesis fraction χ_R increases, other fractions, like biosynthetic pathways (see question below), have to decrease. This is enforced by the constraint that protein synthesis fractions sum up to unity: $\sum_j \chi_j = 1$, see new Eq. (S5) in the SI Note 1.2, which is now a stand-alone equation and referenced in the main text (line 105). We added a statement on the coordination of the proteome fractions in lines 104 to 105.

2/ The issue related to the metabolism leads to an important question. Indeed, it is known that some enzymes necessary at a low growth rate are no longer necessary in the rich medium. So if we suppose that they are, as the ribosome, also in excess, at low growth rate, this raises the question of their uselessness and their disappearance by 'dilution' (and the fact that they occupy for a while a place in the cytosol).

In SI Note 1.2 and Note 1.3 we discuss the dynamics of proteome fractions. The time evolution of a protein sector, e.g. the dilution of "useless" biosynthetic proteins, is set by Eq. (S8). The time scale for dilution is set by the instantaneous growth rate, see Eq. (S10). We have improved the presentation in these Notes to make them more accessible to the reader.

We want to note that while in steady state, synthesis of "useless" proteins (e.g. catabolic, biosynthetic, ...) will always be deleterious, see Eq. (10), these proteins can be very beneficial in other dynamic environments. We discuss such scenarios at the very end of our manuscript (lines 326 – 353).

3 / Finally, I do not understand why the authors do not position more clearly this work with respect to Dai et al. 2016. Indeed, it constitutes an important premise of this work. Moreover, with regard to the data produced in Dais et al 2016, it seems possible to compare the values of λ_0 obtained experimentally with the value that can be deduced with the data of Bai et al. 2016. In particular, in view of Bai et al., it seems also possible to refine the parameters associated to the model of Scott et al?

In the revised version of the manuscript, we put our work into a clearer position with respect to Dai et al. by adding the following statement (lines 42 – 46):

“Mechanistically, the cell achieves this overcapacity by an inactivation of ribosomes, which decreases the amount of active ribosomes, while keeping the elongation rate of the active ribosomes reasonably high¹⁰; indeed, during slow growth as much as 80% of all ribosomes are in a non-translating state¹⁰. This high degree of overcapacity raises the obvious question of what benefit this overcapacity might provide for the cell.”

Concerning the refining of the parameters, Dai et al 2016 indeed characterize the cell finer than previous works (Scott et al 2011, You et al 2013). The authors find a reduction in elongation rate (from 16 aa/s to 8 aa/s). This knowledge combined with the known R/P line allows deducing a reduction of the active ribosome fraction (from 90% to 20%) for slow growth. For our work, it is sufficient to study the translational activity, i.e. the product of elongation rate and active ribosome fraction, see Eq. (S4) in Supplementary Note 1.1 and the surrounding discussion. Additional microscopic information of how much of the reserve is distributed to a reduction of elongation rate and the active ribosome fraction would not improve the prediction in Fig. 2d. We thus prefer the coarse theoretical description of the current manuscript.

The second part of the paper is from my point of view speculative. The presented arguments and the associated computations do not provide more than expected. The question here is not to say that this part is incorrect, only to say that it provides a rational leading to think that the maximization of the growth rate is not necessarily the best solution with respect to specific ecologic niches... It seems to my view more interesting to point out that the bacteria has to manage the resources with respect to the Scott et al. and by taking into account another dimension of the problem: the necessity to use a part of resources to manage transitory/dynamical phase.... This part of the article then points out to my understanding that this new trade-off is (obviously) strongly dependent on the ecologic niche and the considered scenario ...

We are glad to hear that the Reviewer appreciates the second part of the manuscript, despite its more speculative nature. We adjusted the discussion concerning the ecology slightly, and point out trade-off and its strong ecological dependence. In particular, in the discussion we added the statement:

“As a result, we found that cell growth is constrained by a fundamental trade-off between maximization of steady state exponential growth and rapid growth recovery after nutrient upshift. The optimal solution, i.e. the reserve size that maximizes overall growth, strongly depends on the ecological niche. In the specific case of famine-to-feast transition studied here, the ecological niche is characterized by the feast time T , and the optimal overcapacity depends on T according to Eq. (11).” (lines 300 to 305)

Reviewer #3

The authors have carefully addressed the previous concerns and the manuscript is greatly improved. The experimental validation of the model is an important addition and the derivations are much easier to follow. Regarding the experimental validation, I separate it into two parts. For the steady-state data presented in figure S2 and S6, there seems to be enough data to establish a linear relationship between growth rate and R/P mass ratio. However, the assumption of instantaneous shift of σ and χ is based on previous studies. Nevertheless, how well the model presented in this paper fits the measurements can be seen as a validation of this assumption.

I feel that this analysis is compelling, but some statistical aspects should be improved in order to have a strong case for the model:

We are glad to hear that the Reviewer is satisfied with the changes performed on our manuscript. We address the specific concerns of the reviewer about the statistics below.

- In figure 2b, the authors claim there is substantial scatter due to the time derivative. A proper estimation of the error would be very helpful in this case (visualized as error bars in the plot). Same goes also to figure S4, where the fits are even poorer.

As requested, we now report error bars on the instantaneous growth rate in both Fig. 2b and Fig. S4. Growth rates are computed by taking the slope of the best fitting line passing through three consecutive $\log(M)$ measurements, i.e. a 10 min time range. The resulting standard deviation for this procedure, i.e. the size of the error bar, is $\sigma_\lambda = \sqrt{2}\sigma_r/2\Delta t$, where Δt is the time interval between measurements (5 minutes) and σ_r is the uncertainty of $\log(M)$. We estimate σ_r by taking the standard deviation of the residuals between $\log(M)$ and the fit. In the revised manuscript we describe how the instantaneous growth rates, as well as their errors bars, are computed in the captions of Figure 2 and Figure S4. We want to stress that the growth rates are not used for any fitting.

- The model predicts λ_0 as a function of λ_i (for the experimental setup where λ_f was kept constant). Why does figure 2c show the best fit line and not the model prediction? Instead, figure 2d shows λ_0 as a function of ϕ_{Ri} together with the model prediction. This is a peculiar decision, especially given that ϕ_{Ri} is simply a linear transformation of λ_i using the parameters from figure S2, so that would yield $\lambda_0 = 0.36 + 0.82 * \lambda_i$. I fail to see the point of showing the best fit in Fig2c and its parameters if they are not used anywhere else. Also, if we were to believe the provided fit, that would mean that there would be a zero jump in growth rate when $\lambda_i = 0.46/(1-0.59) = 1.12/h$. This is a very different value than the expected 2.45/h.

The Reviewer is criticizing our presentation of Fig. 2, in particular the choice of showing a linear fit of the data in Fig. 2c, instead of the model prediction. We would like to thank the reviewer for highlighting this flawed aspect of the presentation. This prompted us to rearrange Fig. 2 by switching panels (c) and (d), replaced the linear fit with the model prediction. We further improved the presentation in the main text (in particular lines 148 to 166). We think that the manuscript greatly benefitted by this rearrangement.

In line with the other suggestions by the Reviewer, we included uncertainties in the theoretical predictions, adding confidence bands to the theoretical lines, where we propagated uncertainties of all measurements into the theoretical prediction (mostly stemming from the uncertainty in the final growth rate and R-protein fraction). All data shown in Fig. 2c and 2d lie reasonably well within our theoretical predictions.

- In Figure S3, model fitting was done in linear scale, which is not a natural scale for fitting an exponential function. In this case, a semilog-y plot would not make it linear, but could still help guide the eye. Alternatively, the authors could consider transforming the x-axis using $t \rightarrow \exp(\lambda_f * t) - 1$, which should yield a linear relationship if the model is correct.

In the previous version of the manuscript we did not describe carefully how the fit was performed. We agree that a log-transformed biomass $M(t)$ is natural for fitting; in fact, that was the quantity we are using to fit λ_0 (Fig. 2a and S3 are in semilog-y scale). We fit the theoretical prediction for biomass $M(t)$ to optical density by minimizing the sum of the squared difference of the measured $\log M(t)$ and the model (the logarithm of Eq. (7)). We added a statement to clearly describe our fitting procedure in the revised version of the manuscript in Table S2, where the values of λ_0 are presented.

- From looking at figure S4, it looks like there should be large uncertainty in the value of λ_0 and that should also be reflected in figure 2c-d as error bars.

Effectively, Figure S4 might suggest that λ_0 is affected by large errors. However, λ_0 is fitted to $\log(M)$ (discussed above), which constrains λ_0 much more tightly than the instantaneous growth rate. Figure S4 is deceptive, since the small noise present in Fig. S3 is amplified by taking the discrete time derivatives over a short time period (10 min).

The uncertainty in λ_0 , σ_{λ_0} , stems from two sources, one being the scatter of the experimental values of $\log(M(t))$, σ_M , and the other being the error in the final growth rate, which we estimate to be $\sigma_{\lambda_f} = 0.2/h$. The former term can be estimated by jack-knife resampling, i.e. fitting λ_0 by neglecting one biomass measurement at time and studying the dispersion of the fit results. The total error is then obtained by error propagation:

$$\sigma_{\lambda_0} = \sqrt{\sigma_M^2 + \left(\frac{d\lambda_0}{d\lambda_f}\right)^2 \sigma_{\lambda_f}^2}$$

The values of $d\lambda_0/d\lambda_f$ range from -0.2 (for aspartate) to about -0.5 (for glucose). The dominant source of error turns out to be the uncertainty in the final growth rate. Overall, we obtain errors σ_{λ_0} between 0.06 and 0.10/h, which are now reported in Fig. 2c and 2d as error bars and in Table S2. We now describe the uncertainty calculation in the caption of the Table S2.

Minor comments:

Line 169: "overcapacitie" → overcapacity

Line 187: "defined as as" → remove one 'as'

Line 206: "at a givern feast time" → given

Line 219: "jump in growth rate at the from poor to rich" → remove 'at the'

Line 230: "effect on stead state" → 'steady state'

These typographic errors were corrected in the revised manuscript.

REVIEWERS' COMMENTS:

Reviewer #2 (Remarks to the Author):

Sorry for the delay.

The new version takes into account a part of my main critics and so I would like to thank the authors for that.

Clearly, I do not fully agree with some authors' perspective. In particular and to my view, they underestimate too much the role of the regulatory network in the observed behavior: this quite complex regulatory network realizes a fine coordination of the cell components production in order to "saturate the constraints" imposed by the allocation of resources...

In this context, even if most of the "metabolites are abundant", the study of proteomic data shows that the cell proteome is strongly modified when the growth rate increases and so that requires a quite smart adaptation of the cell program ...and not just only the modification of the ribosome abundance

However, it is also clear that is just a problem of perspective and so it is clearly not a reason for me to reject a paper...

Minor comments:

1/ In few sentence, the authors said that the bacteria "intentionally" or to some extent "optimize" Even if I understand well this way to write the sentences (and to short cut a long sentence around the natural selection..), I know that many readers can be "choked" by such sentences .. So I think that it is better to modify such sentences in order to indicate that in some way that "evolution selects such a behavior"

2/ Since the authors cited a large set of papers on the resource allocation, some references are clearly missing on this context, in particular one of Goelzer et al 2015 on *B. subtilis* where the authors has developed and experimentally tested a resource allocation model at the genome scale.

Reviewer #3 (Remarks to the Author):

The authors have carefully addressed all of my comments and I'm happy to recommend this manuscript for publication.

Reviewer #2

Sorry for the delay.

The new version takes into account a part of my main critics and so I would like to thank the authors for that. Clearly, I do not fully agree with some authors' perspective. In particular and to my view, they underestimate too much the role of the regulatory network in the observed behavior: this quite complex regulatory network realizes a fine coordination of the cell components production in order to "saturate the constraints" imposed by the allocation of resources... In this context, even if most of the "metabolites are abundant", the study of proteomic data shows that the cell proteome is strongly modified when the growth rate increases and so that requires a quite smart adaptation of the cell program ...and not just only the modification of the ribosome abundance

However, it is also clear that is just a problem of perspective and so it is clearly not a reason for me to reject a paper...

We thank the Reviewer for being critical yet fair in his/her response. The fine coordination of the proteome is indeed interesting, and not much is known about how it changes in response to nutrient shifts.

Minor comments:

1/ In few sentence, the authors said that the bacteria "intentionally" or to some extent "optimize" Even if I understand well this way to write the sentences (and to short cut a long sentence around the natural selection..), I know that many readers can be "choked" by such sentences ..

So I think that it is better to modify such sentences in order to indicate that in some way that "evolution selects such a behavior"

In response to the Reviewer we changed the following formulations with respect to our use of the words "optimal" and "intentionally" (strikethrough: deleted, underlines: new addition):

Line 71 (previously first occurrence of "optimal"): "Our theory allows us to compute the ~~optimal~~ amount of this reserve that is expected to be selected evolutionarily, i.e. the amount because it maximizes the overall biomass accumulated throughout the course of the shift, for different duration of feast time and growth medium."

Line 252 (now first occurrence of "optimal"): "We also show as a red line the optimal overcapacity $\phi_{R0}^{opt}(T)$, i.e. the overcapacity that maximizes the fitness W for the corresponding feast time T . Strains with this overcapacity $\phi_{R0}^{opt}(T)$ are expected to be selected evolutionarily for re-occurring famine and feast cycles of feast time T ."

Line 330: "This overcapacity, which is substantial at low growth rates¹⁰, is suggested to be a reserve; ~~intentionally kept by the cells to that~~ enables rapid growth upon improvement in nutrient conditions."

Line 336: The optimal solution of the reserve, i.e. the reserve size that i.e., the one expected to be selected evolutionarily because it maximizes overall growth, would depend strongly on the properties of the ecological niche. In the specific case of famine-to-feast transition studied here, the ecological niche is characterized by the feast time T , and the optimal overcapacity depends on T according to Eq. (11)

We kept the word optimal in the context of the mathematical results section (Lines 247 to 317). In the beginning of this section we now clearly define our definition of the word "optimal", see the change mentioned above (Line 252).

2/ Since the authors cited a large set of papers on the resource allocation, some references are clearly missing on this context, in particular one of Goelzer et al 2015 on *B. subtilis* where the authors has developed and experimentally tested a resource allocation model at the genome scale.

We include the reference requested by the Reviewer. In addition, we felt that citing the two following works could further benefit the discussion of the proteome allocation:

29. Weiße, A. Y., Oyarzún, D. A., Danos, V. & Swain, P. S. Mechanistic links between cellular trade-offs, gene

- expression, and growth. *Proc. Natl. Acad. Sci.* **112**, E1038–E1047 (2015).
31. Maitra, A. & Dill, K. A. Bacterial growth laws reflect the evolutionary importance of energy efficiency. *Proc. Natl. Acad. Sci. U. S. A.* **112**, 406–11 (2015).

Reviewer #3

The authors have carefully addressed all of my comments and I'm happy to recommend this manuscript for publication.

We thank the Reviewer for the support of our work, and the critical comments that lead to the current version of the manuscript.